# Rumen Microbiome Reveals the Differential Response of CO_2_ and CH_4_ Emissions of Yaks to Feeding Regimes on the Qinghai–Tibet Plateau

**DOI:** 10.3390/ani12212991

**Published:** 2022-10-30

**Authors:** Qian Zhang, Tongqing Guo, Xungang Wang, Xiaoling Zhang, Yuanyue Geng, Hongjin Liu, Tianwei Xu, Linyong Hu, Na Zhao, Shixiao Xu

**Affiliations:** 1Northwest Institute of Plateau Biology, Chinese Academy of Sciences, Xining 810008, China; 2University of Chinese Academy of Sciences, Beijing 100049, China

**Keywords:** feeding regimes, functional genera, greenhouse gas, rumen microbiota, yak

## Abstract

**Simple Summary:**

Yaks are one of the sources of greenhouse gas emissions from livestock farming on the Qinghai–Tibet Plateau region, and regulating greenhouse gas emissions from yaks has important ecological significance. In this study, we evaluated potential links between basal diet, rumen microbiota composition, and CH_4_ and CO_2_ emissions of yaks under different feeding regimes. We found the CO_2_ and CH_4_ emissions of yaks were lower in traditional grazing than in warm-grazing and cold-indoor feeding regimes. The rumen microbiota of the yaks changed because of differences in basal diet. The CO_2_ and CH_4_ emissions of yaks were related to complementarity among members of the rumen functional genera. We believe that shifts in feeding regimes are effective measures reducing greenhouse gas emissions from yaks and rumen microbiome characterization could be useful screening tools for selecting yaks with low gas emissions.

**Abstract:**

Shifts in feeding regimes are important factors affecting greenhouse gas (GHG) emissions from livestock farming. However, the quantitative values and associated drivers of GHG emissions from yaks (*Bos grunniens*) following shifts in feeding regimes have yet to be fully described. In this study, we aimed to investigate CH_4_ and CO_2_ emissions differences of yaks under different feeding regimes and their potential microbial mechanisms. Using static breathing chamber and Picarro G2508 gas concentration analyzer, we measured the CO_2_ and CH_4_ emissions from yaks under traditional grazing (TG) and warm-grazing and cold-indoor feeding (WGCF) regimes. Microbial inventories from the ruminal fluid of the yaks were determined via Illumina 16S rRNA and ITS sequencing. Results showed that implementing the TG regime in yaks decreased their CO_2_ and CH_4_ emissions compared to the WGCF regime. The alpha diversity of ruminal archaeal community was higher in the TG regime than in the WGCF regime. The beta diversity showed that significant differences in the rumen microbial composition of the TG regime and the WGCF regime. Changes in the rumen microbiota of the yaks were driven by differences in dietary nutritional parameters. The relative abundances of the phyla Neocallimastigomycota and Euryarchaeota and the functional genera *Prevotella*, *Ruminococcus*, *Orpinomyces*, and *Methanobrevibacter* were significantly higher in the WGCF regime than in the TG regime. CO_2_ and CH_4_ emissions from yaks differed mainly because of the enrichment relationship of functional H_2_- and CO_2_-producing microorganisms, hydrogen-consuming microbiota, and hydrogenotrophic methanogenic microbiota. Our results provided a view that it is ecologically important to develop GHG emissions reduction strategies for yaks on the Qinghai–Tibet Plateau based on traditional grazing regime.

## 1. Introduction

The atmospheric concentration of greenhouse gases (GHGs) has increased steadily in the last few decades [1,2], at the same time, they have a significant impact on global warming. Global livestock farming contributes significantly to GHG emissions, generating approximately 14.5% of total global GHG emissions [3,4]. As a result of population growth, rising incomes, and changes in lifestyles and diets, the global demand and production of livestock products are increasing rapidly [5], especially in the developing world, which is possibly further accelerating global GHG emissions [6,7,8]. To meet the Paris Agreement goal of keeping global temperature rise to well below 1.5 °C above preindustrial levels [9], there is a need to balance the livestock farming keeping for nutrition, health, and well-being, with the urgent need to reduce GHG emissions to deal with the climate crisis [10,11].

The Qinghai–Tibet Plateau is the most important livestock areas in China. In recent decades, the expansion of livestock farming and relatively backward technological means of livestock production have exacerbated GHG emissions [12,13], which has caused widespread concern in China. The government is actively promoting the reduction of GHG emissions from the livestock farming on the Qinghai–Tibet Plateau [14,15]. The yak living on the Qinghai–Tibet Plateau is an important ruminant and supports the development of the livestock farming. However, yaks are also one of the largest contributors to GHG emissions from livestock farming on the Qinghai–Tibet Plateau [16,17]. Therefore, GHG emissions from yaks should be investigated.

Under most environmental conditions, feeding management is the most feasible strategy for reducing GHG emissions from livestock farming [18,19,20]. Currently, most of the effects of feeding management on GHG emissions from cattle production systems have been analyzed using models, an approach that has the advantage of allowing estimates at different scales [21,22,23,24]. These studies have shown consistent results that feeding management can affect GHG emissions by influencing the dietary conditions and habitat of cattle [25,26]. The basis of these model estimates requires empirical data to support them, and in previous studies, region specific emission factors are lacking, therefore having an enormous impact on model estimation results [27,28,29]. This knowledge gap may limit our understanding of the feedback of livestock development on the global climate, especially in the Qinghai–Tibet Plateau region, where livestock development of yaks is dominant. Therefore, accurate measurements of GHG emissions from yaks on the Qinghai–Tibet Plateau is necessary to inform the estimation of GHG emissions from livestock farming in the region.

Most of the GHGs are produced by their rumen fermentation in ruminants, which efficiently breaks down plant biomass and its complex dietary carbohydrates into soluble sugars, which are subsequently converted into nutrients and metabolites usable by host animals, as well as expelling CO_2_ and CH_4_ [30,31,32]. Bacteria, fungi, and archaea in the rumen play a key role in this process by interacting to form a complex mixture of microorganisms [33]. Studies have reported that molecular hydrogen (H_2_) and carbon dioxide (CO_2_) is produced during carbohydrate fermentation by bacteria and fungi and is primarily consumed by archaea in the rumen, thereby metabolizing the production of GHGs [34,35,36]. Changes in feeding management are usually accompanied by changes in dietary nutrition levels and environmental conditions, and these changes may individually or interactively affect the rumen microbiome in ruminants [37,38]. For instance, different diets have different rumen methanogenic community characteristics [39], Succinivibrionaceae with implicated in lower CH_4_ emissions from starch-containing diets [40], and fiber-based diet leads to increased relative abundance of Prevotellaceae, which was closely related to GHG emissions [41]. In summary, deterministic effects driven by diet shape the composition and function of the rumen microbiota, which in turn affects GHG emissions. Therefore, studies on changes in the rumen microbiota of yaks after changes in feeding management can help us understand the differences in their GHG emissions.

In this study, we aimed to explore (i) the emissions and differences in CH_4_ and CO_2_ emissions in yaks under different feeding regimes, (ii) differences in the rumen microbiota of yaks under different feeding regimes, and (iii) the potential mechanisms by which changes in rumen microbiota regulate CH_4_ and CO_2_ gas emissions of yaks. Overall, this study aimed to provide a reference for reducing GHG emissions in the livestock farming on the Qinghai–Tibet Plateau.

## 2. Materials and Methods

### 2.1. Experimental Animal Management

An experiment was performed in pastoral areas of Qinghai–Tibet Plateau (3004 m; 37°55′ N, 100°57′ E), Qinghai Province, China. The study period was from May 2021 to May 2022, in the warm season (May to October) and the cold season (November to April). In May 2021, twelve female yaks of similar weight and morphological characteristics at approximately 2 years of age were randomly divided into two feeding regimes: six yaks in the traditional grazing regime (warm-season grazing + cold-season grazing, TG) and six yaks in the warm-grazing and cold-indoor feeding regime (warm-season grazing + cold-season indoor feeding, WGCF). In the warm season, yaks in both TG (n = 6) and WGCF (n = 6) regimes were managed by grazing in natural pastures, and three yaks in each of the two feeding regimes were randomly selected to form a warm-season grazing group (YWG, n = 6), taking into account the experimental conditions and feasibility. In the cold season, yaks in the TG regime were managed in natural pasture for the cold-season grazing group (YCG, n = 6). yaks in the WGCF regime were managed indoors for feeding for the cold-season feeding group (YCF, n = 6). The diet with forage to concentrate ratios (60:40) were fed to yaks of the indoor feeding, which is the most common feeding strategy in the region. The ingredients and nutrient level of concentrated feed in Table 1. The yaks had access to water throughout the experimental period.

### 2.2. Measurement of GHG Emissions

The GHG emissions of the yaks were measured outside the feedlots of the YCF group; thus, the yaks had to be transported from the grazing site to the measurement site for the YWG and YCG groups, but for the YCF group, the yaks were not transported. The GHG emissions were measured during the warm season, July, August, and September, and the cold season, December, January, and February, using the static breath chamber method [42,43]. The GHG measurement system of the static breathing chamber method consisted of three parts: first, a lifting system consisting of a lifting frame and lifting guide chain for lifting the sealed outer cage cover of the static breathing chamber; second, a measurement system consisting of an iron base (2.50 m × 1.90 m), a sealing tank (welded to the iron base, 0.10 m wide), an iron metabolic cage (2.20 m × 1.50 m × 1.65 m) the sealing outer cage (2.40 m × 1.80 m × 1.80 m), four small fans (fixed on the top of the iron metabolic cage used to mix the gas inside the chamber), two thermometers (fixed on the side of the iron metabolic cage used to record the temperature inside the chamber), one food tank (to ensure diet), one water tank (to ensure drinking water) during the test period in the sealing tank filled with absorbent sponge or filled with water, ensuring the hermetic seal between the sealing tank and the sealing outer cage; and third, a gas analysis system consisting of the Picarro G2508 gas concentration analyzer, which is a real-time analytics system that measures the concentrations of gases along with CO_2_ and CH_4_. On the basis of real-time data analysis and yak condition observation, the GHG emissions of yaks were determined every 2 h during the day for 20 min for a total of 12 measurements. Measurements were taken at the same time of the month for each measurement group, and a 5-day acclimatization test, including transport and static breathing chamber acclimatization, was conducted prior to the measurements. The yak diet materials were kept constant during measurement. The YWG and YCG groups used mowing methods to obtain natural pasture, while the YCF group obtained ration directly from the feedlot and had free access to ration and water throughout the trial period. The overall measurement process is illustrated in Figure 1. The GHG emissions of yak were calculated using a linear least-squares fit to all points in the time series of the gas concentration:

Yaks’ GHG fluxes (*F*, g head^−1^ h^−1^) were calculated using the following equation [42]:(1)$$F=\frac{dc}{dt}\times \frac{V}{N}\times \frac{M}{22.4}\times \frac{P}{P_{0}}\times \frac{273}{273+T}$$
where *dc*/*dt* is the rate of change in GHG concentration in the chamber over time (ppm s^−1^), gas samples that showed linear GHG concentration shift with time (*dc*/*dt*) with R^2^ ≥ 0.9 were used for further analysis; *V* is the volume of the chamber (in m^3^); *N* is the number of yaks; *M* is the molecular weight of GHGs (CO_2_: 44 and CH_4_: 16); 22.4 is the molar volume of gas at standard temperature and pressure (in 1 mol^−1^); *P* is the air pressure in the chamber (in Pa); *P*_0_ is the atmospheric pressure at standard conditions, 1.013 × 10^5^ Pa; *T* is the temperature in the closed static chamber (in °C).

Annual, seasonal, and day cumulative yak GHG emissions were calculated using the following equation [44]:(2)$$E=\overset{n}{\underset{i=1}{\sum }}\frac{f_{i}{+f}_{i+1}}{2}{\times (t}_{i+1}-t_{i}),$$
where *E* is the annual or seasonal cumulative CO_2_, CH_4_ (kg head^−1^) emissions or the day cumulative CO_2_, CH_4_ (g head^−1^ d^−1^), *f* represents the emission of CO_2_ (g head^−1^ h^−1^), CH_4_ (g head^−1^ h^−1^), *i* is the measurements *i*th measurement, (*t*_*i*+1_ − *t_i_*) is the interval between measurements.

### 2.3. Dietary Collection and Nutritional Quality Determination

The diet was collected during each experimental period. The natural pasture was sampled using sample squares (0.5 m × 0.5 m) randomly thrown in the center of the grazing pasture, with 10 samples collected at a time in which the pasture was cut to approximately 2 cm above the ground and composed of a mixture of young, mature, and dry pastures to mimic the feeding behavior of cattle [45]. The indoor feed rations were collected from 10 random 50 g samples. All collected diets were oven dried at 60–70 °C for 48 h and individually ground with a grinder to pass a 1 mm sieve for nutritional quality determination. Natural pasture and indoor ration samples were mixed separately in warm and cold seasons to determine the nutritional composition of each test group.

Dry matter (DM) and total ash content were analyzed in these samples by drying at 105 °C and 550 °C, respectively; organic matter (OM) was calculated as dry matter minus total ash [46]; crude protein (CP) was quantified using the Kjeldahl method of nitrogen determination [47]; neutral detergent fiber (NDF) and acid detergent fiber (ADF) contents were determined using an automatic fiber analyzer [48].

### 2.4. Ruminal Fluid Collection and Sequencing Analysis

Ruminal fluid samples were collected once during the warm season (August) and once during the cold season (January). Ruminal fluid samples from each animal were collected using a gastric tube the morning before feeding. For sampling, approximately 50 ml of ruminal fluid was extracted by inserting a gastric tube through the mouth and manually suctioning. The fluid was filtered through four layers of muslin, and the filtered ruminal fluid was collected, immediately frozen on dry ice, and stored at −80°C for microbial sequencing.

Total genomic DNA samples were extracted using an OMEGA soil DNA kit (M5635-02) (Omega Bio-Tek, Norcross, GA, USA), in accordance with the manufacturer’s instructions, and stored at −20 °C before further analysis. The quantity and quality of extracted DNAs were examined using a NanoDrop NC2000 spectrophotometer (Thermo Fisher Scientific, Waltham, MA, USA) and agarose gel electrophoresis, respectively.

PCR amplification (15–35 cycles) was carried out in quadruplicate 25 μL eactions using Q5^®^ High-Fidelity DNA polymerase (New England Biolabs, Hitchin, UK). The primer sets 338F (5′-ACTCCTACGGGAGGCAGCA-3′) and 806R (5′-GGACTACHVGGGTWTC-TAAT-3′) for bacteria, ITS5 (5′-GGAAGTAAAAGTCGTAACAAGG-3′) and ITS2(5′-GCT-GCGTTCTTCATCGATGC-3′) for fungi, and 1106F (5′-TTWAGTCAGGCAACGAGC-3′) and 1378R (5′-TGTGCAAGGAGCAGGGAC-3′) for archaea were used to amplify the target regions. Sample-specific 7 bp barcodes were incorporated into the primers for multiplex sequencing. Thermal cycling involved initial denaturation at 98 °C for 5 min, followed by 25 cycles including denaturation at 98 °C for 30 s, annealing at 53 °C for 30 s, and extension at 72 °C for 45 s, with a final extension of 5 min at 72 °C.

PCR products were cleaned and quantitated using the Vazyme VAHTSTM DNA Clean Beads (Vazyme, Nanjing, China). The band at the expected size containing the amplicons was cut and purified using the Quant-iT PicoGreen dsDNA Assay Kit (Invitrogen, Carlsbad, CA, USA). After the individual quantification step, amplicons were pooled in equal amounts, and 2 × 250 bp pair-end sequencing was performed using the Illlumina MiSeq platform with MiSeq Reagent Kit v3 at Shanghai Personal Biotechnology Co., Ltd. (Shanghai, China).

Raw sequence data were filtered and analyzed using QIIME2 (https://docs.qiime2.o-rg/2019.4/tutorials/, accessed on 30 June 2022) [49], and the Dada2 method was used for primer removal, quality filtering [50], denoising, splicing and chimer removal. Database sequence counts in each library were normalized by subsampling to the same sequencing depth per sample. Amplicon sequence variant (ASV) based approach was used for phylotyping [51,52].

### 2.5. Statistical Analysis

One-way ANOVA was performed to examine the data on dietary nutrients and GHG emissions, and Duncan’s test was conducted for multiple comparisons when treatments had differences. The 16S rRNA and ITS (internal transcribed spacer) microbiome sequencing data were statistically analyzed via Kruskal–Wallis, PERMANOVA, and Mantel test [53]. Redundancy analysis (RDA), introduced by Capblancq [54], was used to analyze the correlation between dietary nutrients and rumen microbial communities in this study. Data were statistically analyzed using SPSS24.0 or R software v3.6.1 [55,56], and data were visualized using Origin 2021, ImageGP (http://www.ehbio.com/Cloud_Platform/front/, accessed on 15 July 2022) [57] and Genescloud (https://www.genescloud.cn, accessed on 12 July 2022).

## 3. Results

### 3.1. Dietary Nutrition and CO_2_ and CH_4_ Emissions from Yaks

In the TG regime (YWG+YCG), DM, NDF, ADF, and OM were higher by 0.92%, 11.22%, 36.84% and 1.56% in the YCG group than in the YWG group, respectively, and CP was lower by 56.33%. In the WGCF regime (YWG+YCF), DM, ADF, and OM were higher by 0.79%, 10.24%, and 1.57%, respectively, in the YCF group than in the YWG group. Conversely, NDF was lower by 6.06% in the YCF group than in the YWG group. CP was similar in the two groups. Based on the overlap between the two feeding regimes, the difference in the YCG and YCF groups was equivalent to that between the TG and WGCF regimes. Between feeding regimes (YCG vs. YCF), NDF and ADF were higher by 18.40% and 24.13%, respectively, while CP was lower by 56.11%, and DM and OM showed no differences in the YCG than in the YCF groups (Table 2).

In the TG regime (YWG+YCG), CO_2_ and CH_4_ emissions were 28.59% and 87.67% higher, respectively, in the YWG than in the YCG groups. In the WGCF regime (YWG+YCF), CO_2_ emissions were higher by 24.88%, while CH_4_ emissions were lower by 18.06% in the YCF group than the YWG group. Between feeding regimes, CO_2_ and CH_4_ emissions were higher by 26.50% and 18.69%, respectively, in the WGCF than in the TG regimes (Table 3).

### 3.2. Rumen Microbiome Structure of Yaks

Sparse analysis showed that our population captured most of the rumen microbiota from each rumen digestive fluid sample (Appendix A). By quality control and leveling (leveling depth set to 95% of the minimum sample sequence volume) [49], a total of 2,427,106 raw reads and 1,206,851 high-quality sequences were obtained for bacteria, 2,561,329 raw reads and 1,935,483 high-quality sequences for fungi, and 2,524,601 raw reads and 1,791,387 high-quality sequences for archaea in the 18 sample sequences. The statistical analysis of the ASV tables after flat sampling, revealed that the rumen microbial communities of yaks changed in the different experimental groups.

The variation in the alpha and beta diversity of the rumen microbiota was analyzed to compare the response of the rumen microbiota to changes in the two feeding regimes. In the TG regime (YWG+YCG), the alpha diversity between the YWG and YCG did not vary significantly in the bacterial, fungal, and archaeal communities (*p* > 0.05; Figure 2a–c). In the WGCF regime (YWG+YCF), the alpha diversity of the YWG was significantly higher than that of the YCF in the bacterial and archaeal communities (*p* < 0.05; Figure 2a,c), whereas the YWG was significantly lower than that of the YCF in fungal communities (*p* < 0.05; Figure 2b). Between feeding regimes (YCG vs. YCF), the alpha diversity of the YCG was significantly higher than that of the YCF in the archaeal communities (*p* < 0.05; Figure 2c), but the alpha diversity between the YCG and YCF did not vary significantly in the bacterial and fungal communities (*p* > 0.05; Figure 2a,b).

Unconstrained principal coordinates analysis (PCoA) based on Bray–Curtis distance showed that rumen bacterial, fungal, and archaeal communities were significantly clustered in all three experimental groups (*p* < 0.001) (Figure 2d–f). In other words, the rumen microbiota was significantly clustered within the TG (YWG vs. YCG) and WGCF (YWG vs. YCF) regimes, and between the TG and WGCF regimes (YWG vs. YCF).

Overall, at the phylum level, bacterial communities were predominated by Firmicutes (47.39–49.14%), Bacteroidetes (44.88–48.26%), and Proteobacteria (1.16–3.87%); fungal communities were predominated by Ascomycota (22.70–67.74%), Basidiomycota (15.94–31.48%) and Neocallimastigomycota (0.21–7.79%); and archaeal communities were predominated by Euryarchaeota (99.98–100.00%) (Figure 3a–c). Under the TG regime (YWG+YCG) an increase in the relative abundance of Ascomycota (*p* < 0.05) was observed in the YCG group (Appendix A). In the WGCF regime (YWG+YCF), an increase in the relative abundance of Neocallimastigomycota (*p* < 0.01) and Euryarchaeota (*p* < 0.05) was observed in the YCF group (Appendix A). Between the feeding regimes (YCG vs. YCF), the relative abundance of Neocallimastigomycota (*p* < 0.01) and Euryarchaeota (*p* < 0.01) increased in the YCF group, and the relative abundance of Ascomycota (*p* < 0.05) decreased (Appendix A).

### 3.3. Relationships between Dietary Nutrition and Rumen Microbial Communities

Changes in dietary nutrition parameters were key factors driving the succession of rumen bacterial, fungal, and archaeal communities (Mantel test R = 0.385 (*p* = 0.001), R = 0.640 (*p* = 0.001), and R = 0.250 (*p* = 0.010), respectively) (Appendix A). The results from the best identified CP, NDF, and ADF contents were the prevailing factors that explain rumen microbial community succession. Redundancy analysis (RDA) between nutritional parameters and rumen microbial genus levels depicted the interdependence of nutrition parameters and rumen microbial communities. For bacterial communities, the first two canonical axes explained 21.87% and 9.57% of the changes, respectively. *Succiniclasticum* was positively correlated with CP, whereas *Selenomonas* and *Butyrivibrio* were positively correlated with ADF and NDF, respectively (Figure 4a). For fungal communities, the first two canonical axes explained 35.11% and 24.34% of the changes, respectively. *Leptosphaeria* and *Cystofilobasidium* were positively associated with CP, whereas *Claviceps* were positively associated with NDF and ADF (Figure 4b). For archaeal communities, the first two canonical axes explained 42.12% and 0.01% of the changes, respectively. *Methanosphaera* was positively associated with CP, while *Methanobacterium* and *Methanobrevibacter* were positively related to NDF and ADF (Figure 4c). In the bacterial, fungal, and archaeal communities, CP was negatively correlated with NDF and ADF.

### 3.4. Relationships between GHG Emissions and Functional Microbial Genera

The microbial genera potentially involved in CO_2_ and CH_4_ gas emissions were investigated in greater detail because of their known significance in rumen ecosystem functioning (Appendix A). Based on differential abundance analysis performed in microbial genera putatively capable of gas production, the studied system included members of the bacterial genera *Prevotella*, *Ruminococcus*, *Butyrivibrio*, *Selenomona*, *Fibrobacter*, and *Bacteroides*; the fungal genera *Orpinomyces*, and the archaeal genera *Methanobrevibacter* and *Methanobacterium*. In the TG regime (YWG+YCG), *Bacteroides* exhibited higher relative abundance in the YWG group (*p* < 0.05) (Figure 5f). Although *Selenomonas* was the predominant functional bacterial in the YCG group (*p* < 0.05) (Figure 5d), none of the other functional microbial genera showed significant enrichment (*p* > 0.05). In the WGCF regime (YWG+YCF), in the YCF group, the relative abundance of *Fibrobacter* (*p* < 0.01), *Selenomonas* (*p* < 0.05), *Orpinomyces* (*p* < 0.01), and *Methanobrevibacter* (*p* < 0.05) increased (Figure 5d,e,g,h). in the YWG group, the relative abundance of *Butyrivibrio* (*p* < 0.05), *Bacteroides* (*p* < 0.05), and *Methanobacterium* (*p* < 0.05) increased (Figure 5c,f,i). Between feeding regimes (YCG vs. YCF), the taxa were significantly enriched in the YCF group comprising functional microbial genera from *Prevotella* (*p* < 0.05), *Ruminococcus* (*p* < 0.05), *Orpinomyces* (*p* < 0.01), and *Methanobrevibacter* (*p* < 0.05) (Figure 5a,b,g,h). The taxa were significantly enriched in the YCG group included functional microbial genus from *Butyrivibrio* (*p* < 0.05) and *Methanobacterium* (*p* < 0.05) (Figure 5c,i).

## 4. Discussion

Typically, changes in feeding regimes are accompanied by differences in dietary conditions because feeding regimes are linked to seasonal and human factors [58,59]. These results were consistent with our observations, which showed that the nutritional parameters of the diet varied significantly within and between the feeding regimes. In ruminant feeding, changes in dietary conditions modulate rumen microsystems by regulating the development and colonization of rumen microbiota [38], influencing diversity, composition, and function of rumen microbiota, and interfering with the metabolism and energy absorption of hosts [60,61]. In addition, changes in the rumen microbiota allow ruminants to adapt to changes in dietary conditions [17,62,63]. Our study also found that rumen microbiota was significantly correlated with dietary conditions. The fiber and protein composition of the diet were the key factors influencing the rumen microbiota. The results of covariation in dietary conditions and rumen microbiota caused by shifts in feeding regimes enhanced our confidence in the potential mechanisms explaining the differences in GHG emissions from yaks within and between feeding regimes.

Previous studies reported conflicting results regarding the relationship between feeding regimes and GHG emissions [64,65,66,67]. We found that the CO_2_ and CH_4_ emissions of yaks were lower in the TG regime (YWG+YCG) than in the WGCF regime (YWG+YCF). This result was consistent with previous findings that grazing systems have lower GHG emissions than indoor feeding systems in cattle [65], sheep [66], and that the addition of a high-concentrate diet promotes GHG emissions from beef cattle [62]. In contrast to the results of Ding et al. [67], who found higher GHG emissions in grazing yaks than in indoor feeding yaks, our findings exhibited differences possibly because of methodological and experimental design limitations in a previous work. In our study, the method of GHG measurement was a combination of static breathing chamber and Picarro G2508 gas concentration analyzer, which can accurately measure the GHG emissions of yaks in real-time. During the measurement process, we supplied a wild natural diet for yaks, which can greatly preserve the native microbial community of the yak rumen and improve measurement accuracy [68]. In addition, CO_2_ and CH_4_ emissions from yaks in the cold and warm seasons within different feeding regimes have been rarely investigated. In our study, we found that CO_2_ and CH_4_ emissions were higher in the warm season than the cold season in the TG regime (YWG+YCG). In the WGCF regime (YWG+YCF), CH_4_ emissions were higher in the warm season than in the cold season, whereas CO_2_ emissions were lower than in the cold season. The reason why these results have appeared might be that rumen microbiome of yak cope with seasonal changes in diets, temperature, and environmental factor demands [62,69,70]. We did not detect any obvious evidence of these tradeoffs in yaks, but future work that seasonal temperature and environmental factor data is needed to help determine if such tradeoffs exist.

The rumen of yaks is a stable and extremely complex microecosystem with a complex microbial community that includes numerous bacteria, fungi, and archaea [70]. These microbiota ferment indigestible carbohydrates and derive energy from them to grow and continue to actively produce volatile fatty acids, CO_2_, H_2_, CH_4_, and others [71], the main gases produced by bacteria and fungi are CO_2_ and H_2_, and part of the dissolved H_2_ and CO_2_ is used by methanogenic bacteria to form CH_4_ [36]. Therefore, we will explain these differences in GHG emissions in the analysis of the yak rumen microbiota.

In the TG regime (YWG+YCG), alpha diversity had no differences, but the beta diversity of rumen bacterial, fungal, and archaeal communities significantly changed between the YWG and YCG groups in yaks. At the phylum level, the relative abundance of Ascomycota increased in the YCG group. These results were due to the stable structure and composition of rumen microbiota developed in yaks over a long period of evolution. Similar to our results, Guo et al. [63] found relatively stable changes in the gut microbiota composition in response to changes in diet composition across seasons. This implied that high-altitude mammals have evolved stable systems of gut microbiota composition across seasons. Thus, differences in the GHG emissions of yaks in this regime may be due to differences in the key functional microbiota. Surprisingly, we found that *Bacteroides* was significantly enriched in the YWG group. *Bacteroides* is a group of well-characterized carbohydrate fermenters that produce CO_2_ and H_2_ [72], which are major substrates supporting ruminal methanogenesis and usually observed to be enriched in ruminants with high gas emissions [73]. *Selenomonas* was enriched in the YCG group, a group of bacteria confirmed in most studies to be hydrogenophilic, which are mainly involved in fumarate and nitrate reduction metabolic pathways, which were important H_2_ sinks and can effectively compete with methanogenic bacteria for H_2_ [36]. *Selenomonas* is usually enriched in ruminants with low GHG emissions [58,71]. These results confirmed our findings, which showed that CO_2_ and CH_4_ emissions were higher in yaks in the YWG group than in the YCG group.

In the WGCF regime (YWG+YCF), the alpha diversities of rumen bacteria and archaea were lower in the YCF group than in the YWG group. The beta diversity of yak rumen microorganisms was also significantly different. This result was similar to those of previous studies [59,74]. The YCF group allowed yaks to live in a limited space and lost contact with a complex environment, which in turn hindered the maintenance of rumen microbial diversity [75]. This diversity also fits neutral diffusion limitation theory [76]. In addition, diet is an important factor in rumen microbial communities as the YWG group has more diverse dietary options and consumes more micronutrients than the YCF group condition; consequently, rumen microbial diversity was higher in the YWG group than in the YCF group [63]. At the phylum level, we found that the relative abundances of Neocallimastigomycota and Euryarchaeota increased in yaks in the YCF group. Neocallimastigomycota has a high fiber-degrading ability, which can secrete a series of fiber degrading enzymes and produce H_2_, CO_2_, and other substances, during fermentation [77,78]. Simultaneously, Euryarchaeota uses these metabolites to produce CH_4_ [79]. However, the role of Neocallimastigomycota may be greater than that of Euryarchaeota, which may also explain the higher CO_2_ emissions in the YCF group than in the YWG group in our study [80]. In terms of functional microbiota studies, our results showed that the relative abundances of the functional microbiota *Fibrobacter* and *Orpinomyces* were higher in the rumen of the YCF group with high CO_2_ emissions than in the rumen of the YWG group. *Fibrobacter* uses cellulose as a substrate to produce short-chain fatty acids, H_2_, and CO_2_ [35]. *Orpinomyces*, an anaerobic fungus with a longer life cycle and a more indeterminate (polycentric) growth regime, favors its proliferation in animals grazing fresh forage and can utilize various substrates to produce H_2_, CO_2_, acetate, lactate, and ethanol [81,82], which support our results. In the rumen microbiota of the YWG group with high CH_4_ emissions, the relative abundance of *Methanobacterium*, which is a hydrogenotrophic methanogen, significantly increased [83]; the increased relative abundance of hydrogen-producing *Butyrivibrio* [84] and *Bacteroides* [36] also provided fermentation substrates for *Methanobacterium*, leading to a high CH_4_ expression. These results indicated that the CH_4_-producing metabolic process in the rumen microbiota of the YWG group was superior to H_2_ and CO_2_ metabolic processes, whereas H_2_ and CO_2_ metabolic processes were superior to the CH_4_-producing metabolic process in the rumen microbiota of the YCF group.

Between the TG and WGCF regimes (YCG vs. YCF), the yaks in the YCF group had higher concentrate diets than in the YCG group, thereby reducing the diversity of their microbiota [85]. Dietary fiber intake was positively correlated with microbial diversity and pasture in the YCG group with a higher fiber content [86]. Consistent with our results, our findings indicated that the alpha diversity in the rumen archaeal microbiota of the YCG group was higher than that in the YCF group, whereas the beta diversity in rumen bacterial, fungal, and archaeal microbiota differed. It has been shown that higher CH_4_ emissions are usually associated with lower archaeal diversity, and that higher archaeal diversity facilitates efficient energy use and thus reduces GHG emissions [87]. In addition, the rumen microbiota in the YCF group altered the microbial community at the phylum level, including the abundances of Neocallimastigomycota and Euryarchaeota. They are involved in carbohydrate metabolic conversion and CH_4_ production through various carbohydrate active enzymes [78,79,81]. These results suggested that the YCF group had better-quality dietary conditions that favored the growth of gas production-related microbial taxa. In terms of functional microbiota studies, our results showed that the relative abundances of *Prevotella*, *Ruminococcus*, and *Orpinomyces* were higher in the rumen of the YCF group than in the rumen of the YCG group. *Prevotella* is an important protein and polysaccharide degrading genus, and *Ruminococcus* and *Orpinomyces* are involved in fiber degradation and biohydrogenation in the rumen [16,33,83]. Their enrichment is positively correlated with high GHG emissions in ruminants [35,88,89]. Similarly, *Methanobrevibacter* is the dominant archaebacterium and a typical hydrogenotrophic methanogenic bacterium in the rumen [90]. In our study, *Methanobrevibacter* was enriched in the YCF group. Previous studies demonstrated that *Methanobrevibacter* plays a significant role in rumen CH_4_ production and its relative abundance is positively correlated with CH_4_ emissions [91,92,93]. These changes in functional microbiota explained the phenomenon that CO_2_ and CH_4_ emissions were higher in the WGCF regime than in the TG regime.

## 5. Conclusions

We investigated the differences in CO_2_ and CH_4_ emissions from yaks within and between feeding regimes by examining the characteristics of the diversity, composition, and functional microbiota of the yak rumen microbiota to characterize the potential causes of differences in CO_2_ and CH_4_ emissions. We found that dietary differences in feeding regimes were crucial factors of variations in yak rumen microbiota. Differences in GHG emissions from yaks were attributed to the enrichment relationship of functional H_2_- and CO_2_-producing and hydrogen-consuming microbiota and hydrogenotrophic methanogenic bacteria. The functional microbiota within and between feeding regimes differed, but they belonged to gas-producing, hydrogen-consuming, and hydrogenotrophic methanogenic bacteria. Accordingly, we presumed that strong ecotopic complementarity among members of the rumen microbial community could cause differences in CO_2_ and CH_4_ emissions. Combined with GHG emission measurements, rumen microbiome characterization would be a useful screening tool for selecting yaks with low gas emission.

## Figures and Tables

**Figure 1 animals-12-02991-f001:**
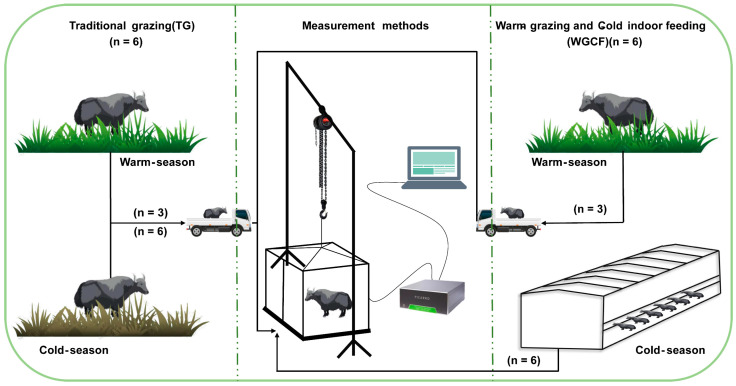
Schematic illustration of the GHG measurement process in two different feeding regimes.

**Figure 2 animals-12-02991-f002:**
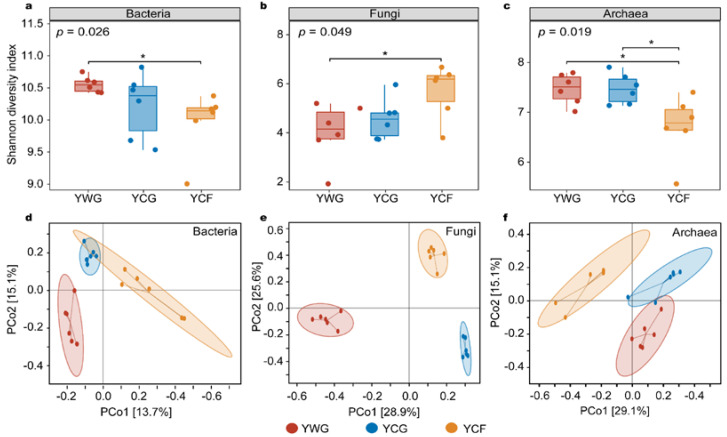
Diversity analysis of rumen microbiota between different experimental groups of yaks. Alpha diversity of the rumen microbiota as determined using the Shannon index. (**a**–**c**) are alpha diversity of the rumen bacteria, fungi, and archaea in the three experimental groups, respectively. Principal coordinates analysis (PCoA) based on Bray–Curtis distances. PCoA analysis of rumen bacteria (**d**), fungi (**e**), and archaea (**f**) composition between different experimental groups. YWG: Warm-season grazing, YCG: Cold-season grazing, YCF: Cold-season indoor feeding. * *p* < 0.05 (Wilcoxon rank-sum test).

**Figure 3 animals-12-02991-f003:**
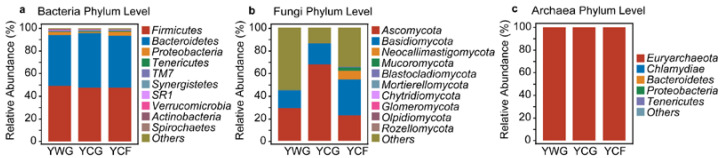
Phylum-level distribution of the rumen bacteria (**a**), fungi (**b**), and archaea (**c**) in three groups. Relative abundance of top 10 for rumen microbiota of yaks. YWG: Warm-season grazing, YCG: Cold-season grazing, YCF: Cold-season indoor feeding.

**Figure 4 animals-12-02991-f004:**
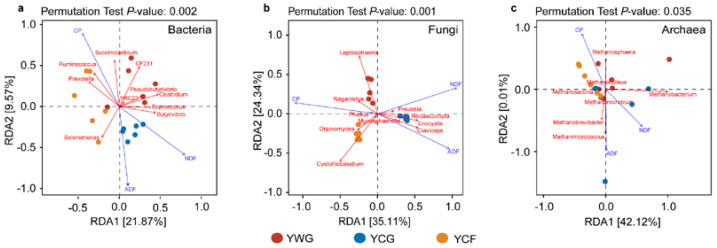
Correlation analysis of the rumen microbial communities with dietary nutrition characteristics. Redundancy analysis (RDA) plot showing the correlations between fermentation characteristics and rumen bacteria (**a**), fungi (**b**), and archaea (**c**) community composition. The canonical axes are labelled with the percentage of total variance explained (%). Arrow lengths indicate the variance explained by dietary nutrition characteristics. Individual yaks from different experimental groups are presented as separate data points. YWG: Warm-season grazing, YCG: Cold-season grazing, YCF: Cold-season indoor feeding.

**Figure 5 animals-12-02991-f005:**
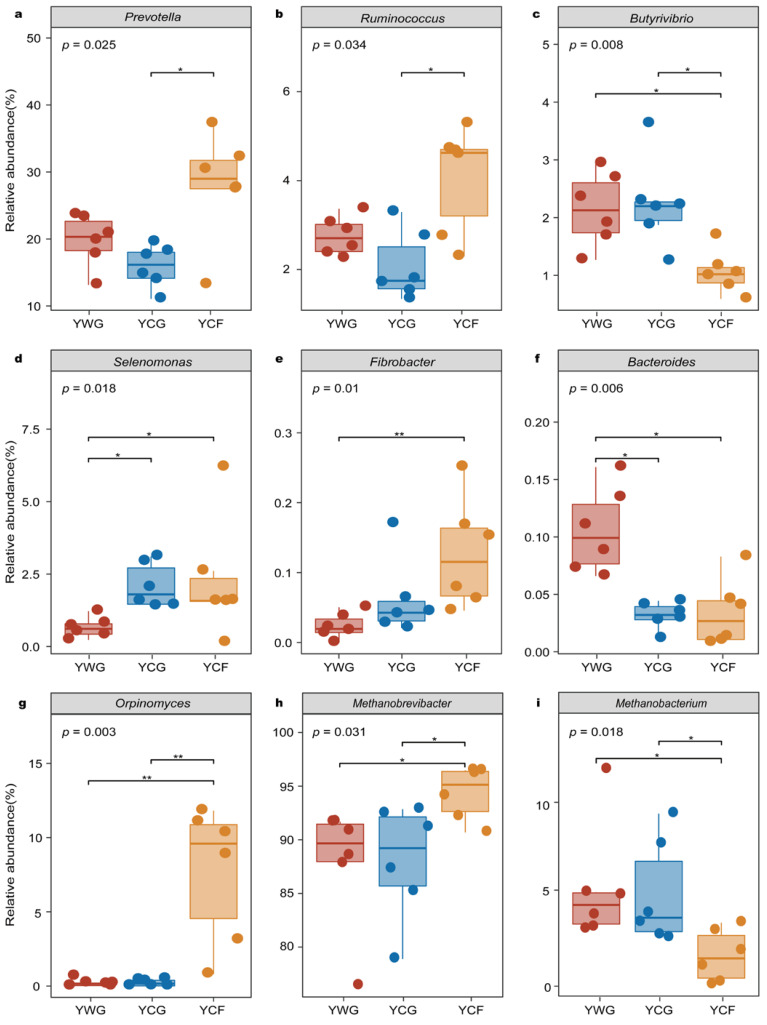
Relative abundance of putative functional bacteria (**a**–**f**), fungi (**g**), and archaea (**h**,**i**) by rumen microbiota compartment and GHG emissions phenotype of yaks. YWG, Warm-season grazing, YWG: Warm-season grazing, YCG: Cold-season grazing, YCF: Cold-season indoor feeding. * *p* < 0.05, ** *p* < 0.01 (Wilcoxon rank-sum test).

**Table 1 animals-12-02991-t001:** The ingredients and nutrient level of concentrated feed (dry matter basis, %).

Items	Ingredients, %	Items	Nutrient Level, %
Oat hay	40.0	DM	83.0
Corn	29.4	CP	11.25
Wheat bran	18.6	NDF	36.70
Rapeseed meal	3.6	ADF	19.31
Corn meal	2.4	Ca	0.69
Soybean meal	3.6	P	0.54
NaCl	0.6		
Premix ^1^	0.6		
CaHPO_4_	0.6		
CaCO_3_	0.6		
Total	100		

^1^ premix (provided per kilogram of complete diet): vitamin A 200,000 IU; vitamin D3 15,000 IU; vitamin E 1250 IU; Cu 375 mg; Fe 15,000 mg; Zn 750 mg; Mn 1000 mg; Se 7.5 mg; DM: Dry matter; CP: Crude protein; NDF: Neutral detergent fiber; ADF: Acid detergent fiber.

**Table 2 animals-12-02991-t002:** Nutrient level and feed intake of experimental groups dietary (dry matter basis, %).

Nutrient Level, %	Groups			SEM	*p*-Value
	YWG	YCG	YCF		
DM	95.81 ^b^	96.69 ^a^	96.57 ^a^	0.08	<0.001
CP	9.96 ^a^	4.35 ^b^	9.91 ^a^	0.49	<0.001
NDF	56.40 ^b^	62.73 ^a^	52.98 ^c^	0.81	<0.001
ADF	27.82 ^c^	38.07 ^a^	30.67 ^b^	0.86	<0.001
OM	91.84 ^b^	93.27 ^a^	93.28 ^a^	0.15	<0.001

YWG: Warm-season grazing; YCG: Cold-season grazing; YCF: Cold-season indoor feeding; DM: Dry matter; CP: Crude protein; NDF: Neutral detergent fiber; ADF: Acid detergent fiber; OM: Organic matter; SEM: Standard error of the mean; Values in the same row with different letters are significantly different (*p* < 0.05).

**Table 3 animals-12-02991-t003:** The day, seasonal, and annual cumulative CO_2_ and CH_4_ emissions from yaks.

Groups	CO_2_ Emissions			CH_4_ Emissions		
	Day, g head^−1^ d^−1^	Seasonal, kg head^−1^	Annual, kg head^−1^	Day, g head^−1^ d^−1^	Seasonal, kg head^−1^	Annual, kg head^−1^
YWG	1729.80 ^b^	311.36 ^b^	-	56.18 ^a^	10.11 ^a^	-
YCG	1345.18 ^c^	242.13 ^c^	-	29.94 ^c^	5.39 ^c^	-
YCF	2160.17 ^a^	388.83 ^a^	-	46.03 ^b^	8.29 ^b^	-
SEM	81.96	14.75	-	2.63	0.47	-
*p*-value	<0.001	<0.001	-	<0.001	<0.001	-
TG	-	-	553.50 ^b^	-	-	15.50 ^b^
WGCF	-	-	700.20 ^a^	-	-	18.40 ^a^
SEM	-	-	22.64	-	-	0.44
*p*-value	-		<0.001	-	-	<0.001

YWG: Warm-season grazing; YCG: Cold-season grazing; YCF: Cold-season indoor feeding; SEM: Standard error of the mean; TG: Traditional grazing; WGCF: Warm-grazing and cold-indoor feeding; Warm season (May to October) and cold season (November to April); Values in the same column with different letters are significantly different (*p* < 0.001).

## Data Availability

All raw sequences for this study can be found in the NCBI Sequence Read Archive under BioProject PRJNA874346 with the accession number SUB11978917 (https://dataview.ncbi.nlm.nih.gov/object/PRJNA874346, accessed on 11 October 2022).

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
