# Peer review of "Rumen Microbiome Reveals the Differential Response of CO2 and CH4 Emissions of Yaks to Feeding Regimes on the Qinghai–Tibet Plateau"

_animals, 2022, doi:10.3390/ani12212991_

Round 1

Reviewer 3 Report

Dear authors

I reading your manuscript I found several interesting cues. In attached some suggestions aimed to improve the original article

Best Regards 

Round 2

Reviewer 1 Report

Dear authors, I would like to carefully review your revised article with the id number "animals-1995926". In my opinion, this article incorporates all of the points raised in the original draft to the fullest extent possible. However, it would be helpful if you could make one last small correction to the text in order to save the readers' eyes from strain. Your tables are quite tiring to read due to the superscript numbers you have provided after the abbreviations. We would greatly appreciate it if you were able to remove all superscript numbers except premix. Aside from this small point, I believe that you have filled a gap that exists in the literature regarding yaks, which are understudied. I wish to congratulate each of the authors who contributed to this beautiful work and wish them continued success in their endeavors.

Author Response

Dear reviewer:

We are grateful for your careful reading and suggestions. We modified the manuscript as the suggestions from the reviewer. The modified words were highlighted in red color in the revised manuscript. We modify these superscript numbers in line 122, 124-125, 249, 250-252, 254, and 255-257 in the revised manuscript.